# Stimuli-responsive hydroxyapatite liquid crystal with macroscopically controllable ordering and magneto-optical functions

Masanari Nakayama [1], Satoshi Kajiyama [1], Akihito Kumamoto[2], Tatsuya Nishimura [1], Yuichi Ikuhara[2], Masafumi Yamato [3] & Takashi Kato [1]

Liquid crystals are mostly formed by self-assembly of organic molecules. In contrast, inorganic materials available as liquid crystals are limited. Here we report the development of liquid-crystalline (LC) hydroxyapatite (HAp), which is an environmentally friendly and biocompatible biomineral. Its alignment behavior, magneto-optical properties, and atomic-scale structures are described. We successfully induce LC properties into aqueous colloidal dispersions of rod-shaped HAp by controlling the morphology of the material using acidic macromolecules. These LC HAp nanorod materials are macroscopically oriented in response to external magnetic fields and mechanical forces. We achieve magnetic modulation of the optical transmission by dynamic control of the LC order. Atomic-scale observations using transmission electron microscopy show the self-organized inorganic/organic hybrid structures of mesogenic nanorods. HAp liquid crystals have potential as bio-friendly functional materials because of their facile preparation, the bio-friendliness of HAp, and the stimuli-responsive properties of these colloidal ordered fluids.

[1] Department of Chemistry and Biotechnology, School of Engineering, The University of Tokyo, 7-3-1 Hongo, Bunkyo-ku, Tokyo 113-8656, Japan. [2] Institute of Engineering Innovation, School of Engineering, The University of Tokyo, 2-11-16 Yayoi, Bunkyo-ku, Tokyo 113-8656, Japan. [3] Department of Applied Chemistry, Graduate School of Urban Environmental Sciences, Tokyo Metropolitan University, 1-1 Minami-ohsawa, Hachioji, Tokyo 192-0397, Japan. Correspondence and requests for materials should be addressed to T.K. (email: kato@chiral.t.u-tokyo.ac.jp)

Liquid crystals form fluid but ordered molecular assemblies[1–3], and these self-assembled structures can be dynamically controlled by application of external stimuli such as electric and magnetic fields, and mechanical forces, leading to various functions[4–11]. Although typical liquid crystals are based on organic molecules with rod and/or disk shapes, colloidal particles with controlled anisotropic morphologies and sizes can also show liquid-crystalline (LC) phases in their dispersed states[12,13] and are expected to have new functional properties[14–20].

The production of functional materials from abundant resources using environmentally friendly processes is important for a future sustainable society[21–23]. Hydroxyapatite (HAp) $[Ca_{10}(PO_4)_6(OH)_2]$ is a major bio-friendly material. As for combination of liquid crystals and HAp crystals, a simple mixture of a LC thermotropic aromatic polyester and non-LC HAp particles was previously reported[24]. However, to the best of our knowledge, there was no report on HAp crystals, which themselves exhibit liquid crystallinity. If HAp could be endowed with LC properties, new bio-friendly and environmentally friendly functional materials could be developed, which might provide interesting materials applications including optical switching, biosensors, and structural materials available for artificial bones, dental implants, and cell culture scaffolds. HAp is one of the main components of human bones and teeth; and the size, morphology, crystallinity, and orientation are biologically well controlled under mild conditions[25]. It has been proposed that acidic proteins transiently stabilize the amorphous phases of these bio-minerals[26,27] as precursors for the formation of sophisticated self-organized structures[28–31].

High external magnetic field has been instrumental in orienting diamagnetic or paramagnetic materials with small anisotropic magnetic susceptibility, such as alumina, titania, or HAp[32–35]. For LC HAp, the use of magnetic fields could be promising for orientational control because self-assembly into LC ordered fluids can promote magnetic alignment[36–38].

Here we report the development of stimuli-responsive liquid crystals based on colloidal nanorod mesogens of HAp (Fig. 1). Our strategy is to use a bioinspired self-organization approach to prepare HAp colloidal liquid crystals. Based on biomineralization processes, we explore the use of acidic macromolecules to synthesize mesogenic HAp nanorods capable of forming colloidal liquid crystals (Fig. 1a–c). The structure of the mesogenic nanorods is clarified by conventional/high-resolution transmission electron microscopy (TEM/HRTEM). The alignment of the LC nanorods using external magnetic fields and mechanical forces (Fig. 1d–f) are also described. Furthermore, our aim is to functionalize HAp liquid crystals through dynamic control of the LC order by application of external magnetic fields. A LC system dynamically responsive to magnetic fields is designed to demonstrate the magneto-optical functions.

## Results

**Structural characterization of HAp-based mesogens.** A colloidal dispersion of HAp was obtained by mixing aqueous $CaCl_2$ and $K_3PO_4$ solutions in the presence of poly(acrylic acid) (PAA) as an additive. Nanorod-shaped HAp crystals were formed, as shown in Fig. 2. The average length and width of the nanorods were $100 \pm 20$ nm and $21 \pm 5$ nm, respectively. The aspect ratio of the nanorods was 5.0. The TEM images (Fig. 2a, c) show that the nanorods had polycrystalline structures comprising rod-shaped nanocrystallites. However, the arched pattern observed in the selected-area electron diffraction (SAED) pattern (Fig. 2b) shows that the polycrystal have the preferred orientation along with the

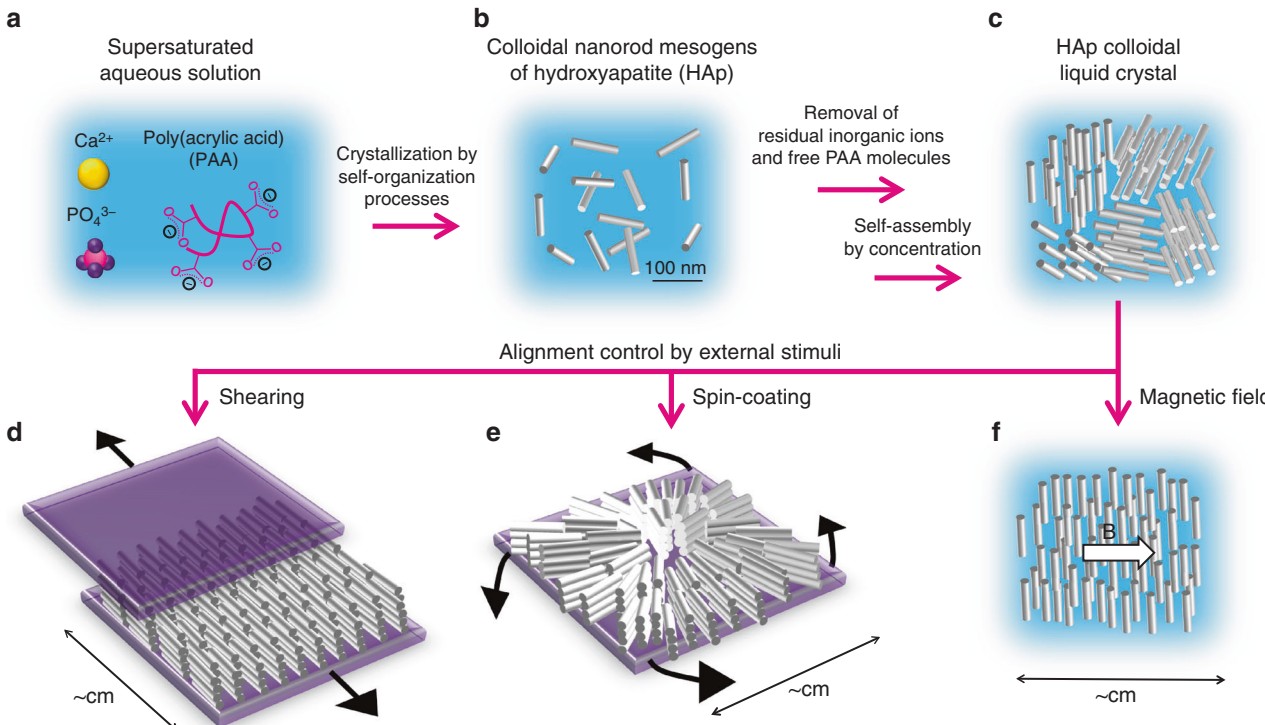

**Fig. 1** Schematic representation of synthesis of HAp colloidal liquid crystal and alignment control by external stimuli. **a** Supersaturated aqueous solution of calcium phosphate in the presence of PAA. **b** Colloidally stable HAp nanorods synthesized using bioinspired crystallization process. **c** LC states of self-assembled HAp nanorods in condensed aqueous colloidal dispersion after removal of residual inorganic ions and free PAA molecules that were not complexed with HAp crystals. **d** Nanorod assembly unidirectionally oriented on a macroscopic scale by mechanical shearing. **e** Nanorod assembly radially oriented on a macroscopic scale by spin-coating. **f** LC aqueous colloidal dispersion showing macroscopic unidirectional alignment in response to a magnetic field

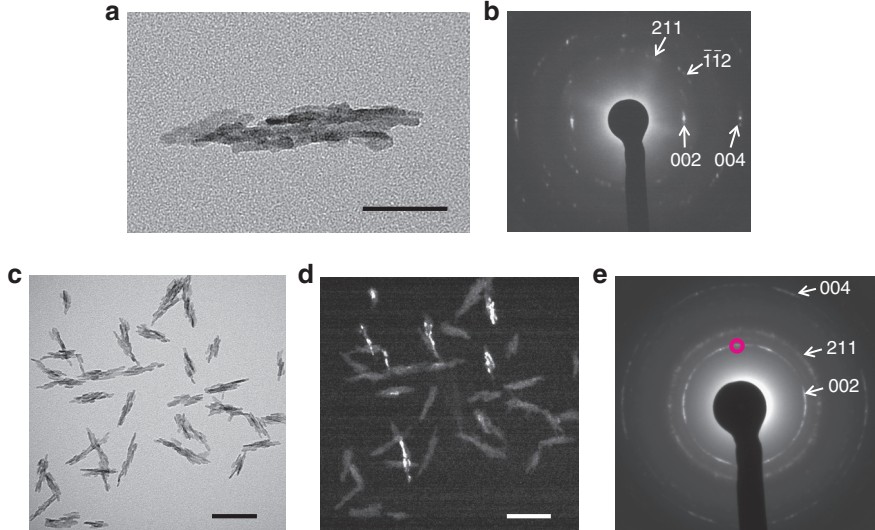

**Fig. 2** TEM analysis of synthesized HAp nanorods. **a** TEM image of one HAp nanorod and **b** corresponding SAED pattern. **c** Bright-field TEM image and **d** corresponding dark-field TEM image of nanorods. **e** SAED pattern corresponding to (**c**). The area of (002) reflection used for the dark-field observation is marked with a magenta circle in (**e**). Scale bars, **a** 50 nm and **c**, **d** 100 nm

$c$ axes of HAp directed to the long axis of the nanorod. The preferred orientation was further analyzed based on the dark-field TEM image (Fig. 2d) corresponding to the bright-field image in Fig. 2c, obtained using the (002) reflection shown by the magenta circle in Fig. 2e. The bright domains of rod-like nanocrystallites were only observed in nanorods with long axes pointing in the direction corresponding to the reflection area in Fig. 2e. These results indicate that the $c$ axes were exclusively aligned along the long axes of the nanorods. X-ray diffraction (XRD; Supplementary Fig. 1a) patterns and Fourier-transform infrared (FTIR) spectra (Supplementary Fig. 2a) show that the precipitates formed in the early stage were amorphous. The amorphous calcium phosphate precursors were spherical, with diameters of around 10–30 nm (Supplementary Fig. 3a). The SAED pattern confirms that the nanoparticles had amorphous-like structure (Supplementary Fig. 3b). Dynamic light scattering (DLS) measurements showed that the average hydrodynamic diameter of the amorphous nanoparticle precursors was $50 \pm 20$ nm (Supplementary Fig. 4), suggesting that these nanoparticles were hydrated in aqueous solution. The measured zeta potentials of the amorphous precursors were $-18 \pm 1$ mV (Supplementary Fig. 5a), indicative of negative surface charges. Thermogravimetric (TG) measurements showed that 17.6 wt% PAA was included in the amorphous precursors (Supplementary Fig. 6a). These results suggest that PAA molecules played significant roles in stabilization of the amorphous and colloidal states of the precursors. The colloidal precursors of amorphous calcium phosphate were transformed into HAp nanorods within 3 days, as shown by XRD and FTIR (Supplementary Figs. 1b and 2b). The formation of HAp nanorods was also confirmed using inductively coupled plasma atomic emission spectroscopy (ICP/AES). The Ca/P ratio in the nanorods was 1.66:1, which is comparable to the stoichiometric value for HAp (1.67:1). Energy-dispersive X-ray spectroscopy (EDS) analysis shows the presence of Ca, P, C, and O elements included in a nanorod, which are all ascribed to HAp, PAA, or water molecules, while no impurities such as $K^+$ and $Cl^-$ were detected (Supplementary Fig. 7). These results suggest high purity of HAp nanorods.

Atomic-level structural information on the nanorods was obtained using a negative spherical aberration imaging technique of HRTEM, properly optimized to be brighter contrast at atom positions[39,40]. The HRTEM images show highly crystalline and anisotropic growth (Fig. 3). The HRTEM image of area A marked in the low-magnification image (Fig. 3a) clearly shows that the nanorods consisted of rod-like nanocrystallites of length around 20 nm, with preferential growth of the $c$ axis along the long axis (Fig. 3b). The nanocrystallite sizes are consistent with those observed in the dark-field image (Fig. 2d). The HRTEM image shown in Fig. 3c corresponds to area B of a nanorod marked in Fig. 3a. Atomic columns can be observed for two different adjacent nanocrystallites, denoted by I and II in Fig. 3c. The zone axis of these nanocrystallites is [320], according to the fast Fourier transform (FFT) patterns (Fig. 3d, e). The experimental image (Fig. 3f) is in good agreement with the simulated image viewed from the [320] direction (Fig. 3g). The orientation directions of the adjacent nanocrystallites are close to each other. One crystallite is connecting to the other via an amorphous organic layer. These results suggest that the assembly of primary particles through oriented attachment is involved in formation of the nanorods. These crystallization processes are widely observed in biomineralization-inspired crystallization mechanisms[41]. The anisotropic mesocrystalline structure[42] of the HAp nanorod is shown in Fig. 3h.

Surface structures of the nanorods were also examined using HRTEM. The results show that the nanorod surface was covered with an amorphous layer of thickness around 1 nm (Fig. 3c, magenta arrows). It is worth noting that the aqueous colloidal dispersion of HAp nanorods was free from aggregation, without surface modification, in spite of the large surface areas. The zeta potential of an aqueous colloidal dispersion of the HAp nanorods was measured (Supplementary Fig. 5b). The value was $-39 \pm 7$ mV, indicating that the surfaces of the nanorods were negatively charged. TG analysis of the nanorods showed that the HAp nanorods were hybridized with 13.7 wt% PAA (Supplementary Fig. 6b). The presence of PAA complexed with the nanorods was also confirmed by direct elemental analysis for C, H, and N elements. The HAp/PAA ratio was estimated to be 5.7:1.0 by weight, which is in good agreement with the value estimated by the TG measurements. From these results, we conclude that PAA molecules complexed with the surfaces of the nanorods, generating electrostatic repulsive forces between the nanorods.

**LC behavior of aqueous colloidal dispersions of HAp nanorods.** Self-assembly of HAp nanorods into liquid crystals was

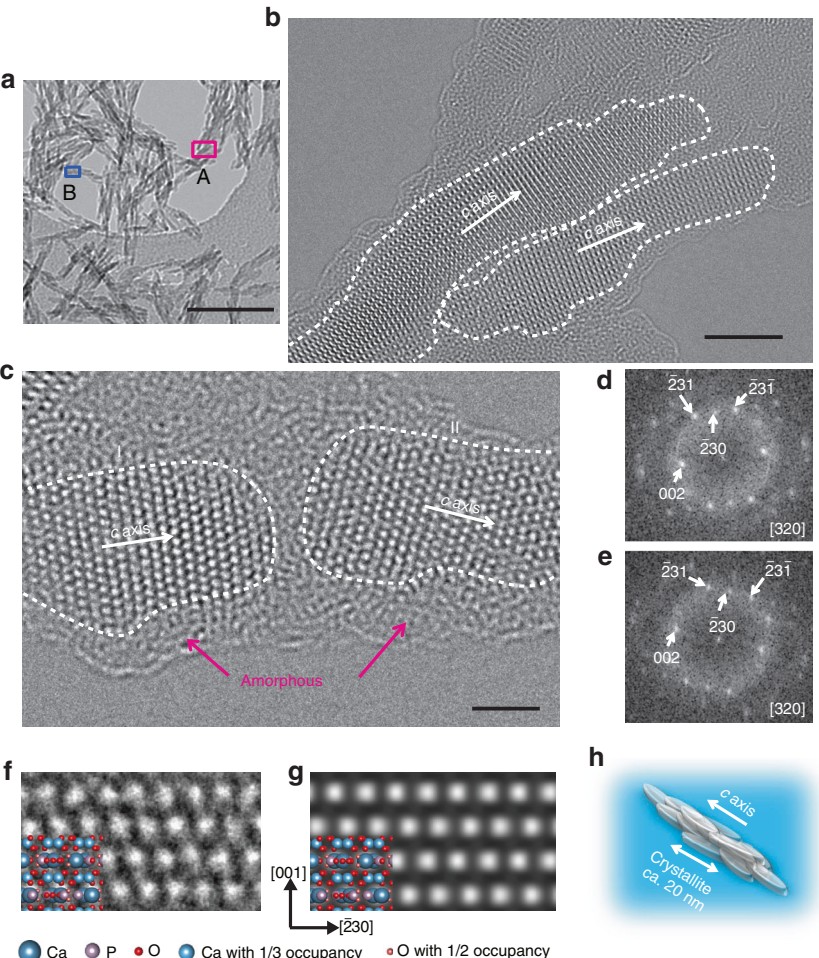

**Fig. 3** HRTEM observations of synthesized HAp nanorods. **a** TEM image of HAp nanorods at low magnification. **b**, **c** HRTEM images of HAp nanorods for areas A and B, respectively, marked in (**a**). **d**, **e** FFT patterns corresponding to the crystallites denoted by I and II, respectively, in (**c**). **f** Enlarged experimental image of crystallite denoted by I in (**c**) with the structure model of HAp viewed from the [320] direction. **g** Simulated image viewed from the [320] direction of HAp with its structure model. **h** Schematic illustration of the mesocrystalline structure of one HAp nanorod. Scale bars, **a** 100 nm, **b** 5 nm, and **c** 2 nm

observed using polarizing optical microscopy (POM; Fig. 4) of the condensed aqueous colloidal dispersions after removal of residual inorganic ions and free PAA molecules that were not complexed with HAp crystals (Fig. 1c). The colloidal dispersions of HAp nanorods were held in glass capillaries at various concentrations. Figure 4a shows the isotropic (Iso)/LC phase separation process at 8.0 vol% with time. In the colloidal dispersion, LC droplets emerged from the Iso phase and gradually settled at the bottom because of their higher density. Iso/LC phase separation was complete within 14 days. A phase diagram of the colloidal dispersions was obtained by measuring the volume ratios of the Iso and LC phases (Fig. 4b). At concentrations below 7.6 vol%, the dispersions only showed the Iso state. The Iso/LC coexisting phase appeared in the volume fraction range 7.6–8.8 vol%. The proportion of the LC phase increased with increasing concentration of HAp nanorods. Colloidal dispersions at concentrations above 8.8 vol% consisted entirely of the LC state (Fig. 4c). The colloidal dispersions condensed to the LC state still maintained sufficient fluidity (Fig. 4d), suggesting potential dynamic properties.

**Macroscopic alignment using mechanical stimuli.** The HAp liquid crystals were macroscopically oriented using external

mechanical stimuli, as shown in Fig. 5. Unidirectional alignment (Fig. 5a–d) of the LC colloidal dispersions was achieved by unidirectional mechanical shearing (Fig. 1d). Radially aligned assemblies (Fig. 5e–h) were obtained by spin-coating (Fig. 1e). The image of the assembly of HAp nanorods oriented by mechanical shearing changed from bright to dark on each rotation of the sample by 45° between crossed polarizers (Fig. 5a, b). Unidirectional alignment was also observed using scanning electron microscopy (SEM) (Fig. 5c). The two-dimensional distribution of the c axes of the HAp nanorods was determined based on the XRD pattern of the nanorod assembly oriented by mechanical shearing (Fig. 5d). The results clearly show that the c axes of HAp were highly aligned along the shearing direction. The orientation degree of the c axes was estimated to be 0.81. Use of a wave plate in the POM observations showed that the unidirectional alignment of these nanorods gives rise to positive birefringence (Supplementary Fig. 8).

Spin-coating of colloidal dispersions with concentrations higher than 7.3 vol% provided a different type of macroscopic scale aligned structure (Fig. 1e). After spin-coating of the colloidal dispersions, the assembly appeared as a crossed birefringence pattern (Fig. 5e). The distribution of the interference color pattern (Fig. 5f) indicates that the c axes of HAp were radially aligned in the assembly, based on the positive birefringence of LC materials

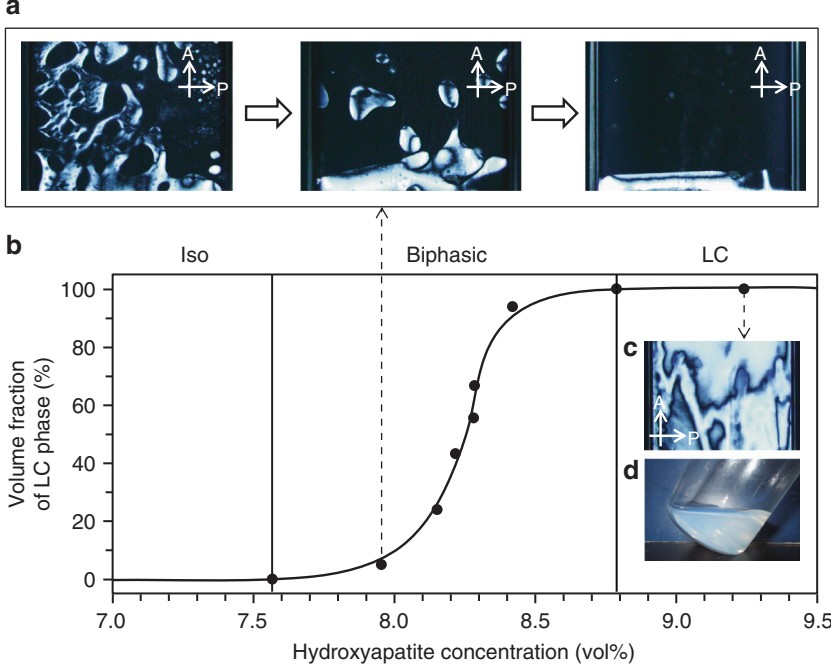

**Fig. 4** LC properties of aqueous colloidal dispersions of HAp nanorods. **a** Iso/LC phase separation for 8.0 vol% aqueous colloidal dispersion of HAp nanorods observed under crossed polarizers after 10 h, 13 h, and 14 days. **b** Phase diagram for aqueous colloidal dispersion of HAp nanorods. **c** Typical textures of HAp aqueous colloidal dispersions forming homogeneous LC state at 9.2 vol% observed using POM. **d** Digital photograph of colloidal dispersion in LC state at 9.2 vol%. A Analyzer, P Polarizer

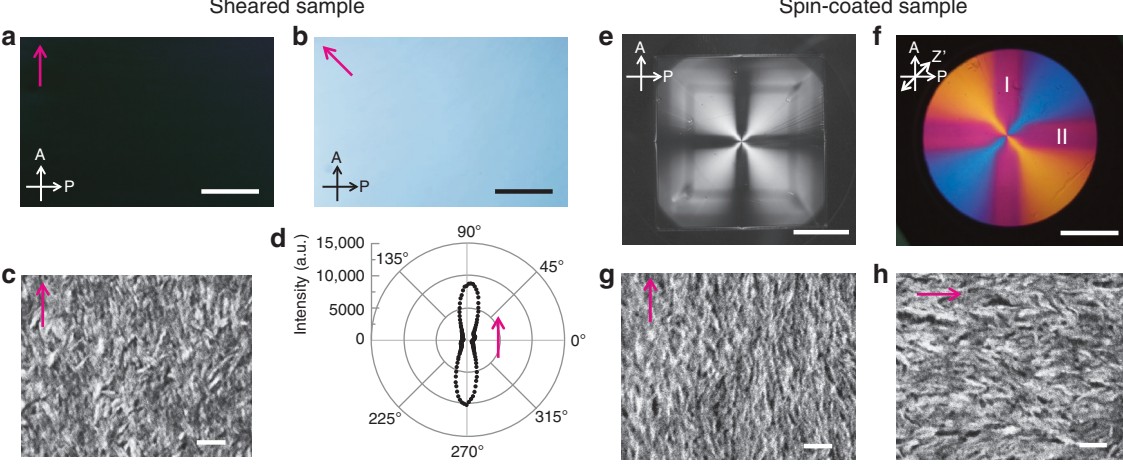

**Fig. 5** Alignment control by mechanical shearing and spin-coating. **a**, **b** Typical POM images and **c** SEM image of assembly of HAp nanorods oriented by mechanical shearing. **d** Two-dimensional distribution of the *c* axes of HAp in the sheared assembly based on XRD. The magenta arrows in (**a–d**) represent the direction of shearing. **e**, **f** Typical POM images obtained (**e**) without and (**f**) with a wave plate of the assembly oriented by spin-coating at 2000 rpm on a glass slide. **g**, **h** SEM images of spin-coated assemblies obtained at positions denoted by I and II in (**f**). The magenta arrows in (**g**, **h**) show alignment directions. A Analyzer, P Polarizer, Z' slow axis of the wave plate. Scale bars, **a**, **b** 1 mm, **c** 100 nm, **e** 1 cm, **f** 0.5 cm, and **g**, **h** 100 nm

(Supplementary Fig. 8). The radial alignment continued seamlessly over an area of 9 cm² (Fig. 5e). For the positions on the coating denoted by I and II in Fig. 5f, the alignment directions were orthogonal to each other (Fig. 5g, h). These results show that the nanorods were aligned from the center toward the periphery to give a macroscopic radial orientation.

**Magnetic alignment of LC colloidal dispersions**. We found that LC colloidal dispersions of the HAp nanorods can be magnetically aligned (Fig. 1f). A magneto-optical response was

successfully achieved for our LC HAp materials under crossed polarizers (Fig. 6). Although the HAp nanorods prepared in the present study are polycrystalline, the nanorods were highly crystalline and crystallographically oriented through the bioinspired self-organization process (Figs. 2 and 3). The nanorods can align in magnetic fields because of crystalline magnetic anisotropy[32,33]. These HAp nanorods in the LC state can be aligned under a 3 T magnetic field (Supplementary Fig. 9). In contrast, the Iso colloidal dispersion of HAp nanorods shows no magnetic response at 3 T and even at 10 T (Supplementary Fig. 10). We investigated the magneto-responsive properties by measuring

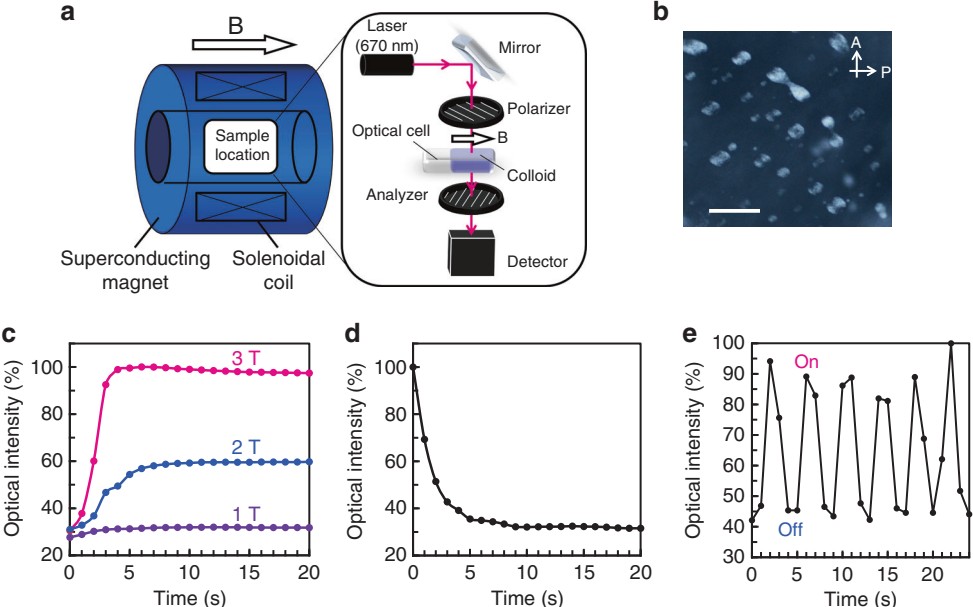

**Fig. 6** Dynamics of magnetic response and relaxation in Iso/LC biphasic aqueous colloidal dispersions. **a** Experimental setup for in situ measurement of light transmission through colloidal dispersions. Polarizer and analyzer were set in the cross-Nicol condition, forming an angle of 45° with the magnetic field direction. The laser wavelength was 670 nm. **b** POM image of 8.5 vol% Iso/LC biphasic colloidal dispersion in 2-mm-thick optical cell. A Analyzer, P Polarizer. Scale bar, 100 μm. **c** Time courses of response of Iso/LC biphasic colloidal dispersion under 1 T, 2 T, and 3 T magnetic fields. **d** Time course of relaxation of Iso/LC biphasic colloidal dispersion after magnetic field of 3 T was turned off. **e** Changes with time of the transmitted light intensity for biphasic Iso/LC colloidal dispersion when the magnetic field of 3 T was repeatedly turned on and off

changes in the transmitted light intensity on application of magnetic fields under crossed polarizers (Fig. 6a). This optical change reflects the change in the orientational order[43]. The HAp nanorods in the LC state required several minutes for the orientational change in the case of both the magnetic response (Supplementary Fig. 11a) and relaxation behaviors (Supplementary Fig. 11b). It is presumed that the response and relaxation processes were hindered by anchoring of the nanorod mesogens on the glass walls of the optical cell[44,45]. This is supported by the slower response speed on the order of hours when the sample thickness was decreased from 2 mm to 0.1 mm (Supplementary Fig. 12).

It is worth noting that the Iso/LC biphasic colloidal dispersions showed an outstanding response to magnetic fields. When the biphasic colloidal dispersion (Fig. 6b) was placed under a magnetic field, it quickly responded to external fields over 1 T and reached the maximum order parameter within a few seconds (Fig. 6c), which is 100 times faster than the response speed of the homogeneous LC colloidal dispersions in the same 2-mm-thick cell (Supplementary Fig. 11a). When the magnetic field of 3 T was turned off, relaxation started immediately, and it only took a few seconds to complete (Fig. 6d); this is 20 times faster than the relaxation speed of the homogeneous LC colloidal dispersions (Supplementary Fig. 11b). This behavior allows rapid oscillations of the transmitted light intensity within a few seconds when the magnetic field of 3 T is alternately turned on and off (Fig. 6e).

The dispersion is highly transparent, with a transmittance of 98% at a wavelength of 670 nm, even in a 2-mm-thick cell (Supplementary Fig. 13). This is because HAp shows no light absorption band in the visible region and the nano-sized mesogens show no light scattering. The optical properties of these LC materials could be used for efficient control of light transmission.

**Magnetic light modulation by biphasic system**. We successfully achieved magnetic modulation of light transmission by dynamic

control of the LC order in the biphasic system (Fig. 7). It should be noted that for the Iso/LC biphasic colloidal dispersion, there was no hysteresis of the optical intensities during increasing and decreasing of the applied magnetic field strength (Fig. 7a). In contrast, the optical intensity of the colloidal dispersion of homogeneous LC states did not return to the initial level after the relaxation process (Supplementary Fig. 11a, b) because of strong anchoring at the liquid crystal/cell wall interface. Reversible switching of the brightness of the biphasic system was also shown by in situ observations under crossed polarizers (Fig. 7b). The optical intensity of the biphasic system precisely traced the changes in the strength of the external magnetic field, as shown in Fig. 7c. In addition, the light transmission was 27 times higher at 10 T than at 3 T (Fig. 7c), showing a large contrast ratio. The dependence of light transmission intensity on the field strength is in close agreement with the theoretical calculation based on orientation change of LC directors induced by magnetic torque (Supplementary Fig. 16). These results suggest that the light transmission can be effectively modulated by tuning the strength of the applied magnetic field. To examine the effects of magnetic field strength on optical modulation, the applied magnetic fields were oscillated between 5 T and 7 T, which led to reversible switching of brightness between two particular values (Fig. 7d). When the applied field strength was changed, the optical transmission was modulated, depending on the field strength (Fig. 7e). Modulation of the transmitted light was also achieved with lower field strengths of around 1–3 T when the intensity of the light source was increased (Supplementary Fig. 14).

## Discussion

In the present work, a family of liquid crystals based on HAp was developed. We successfully obtained magneto-optical functions through control of the LC order of HAp nanorods by external magnetic fields. In general, when a magnetic field is applied to a particle, the magnetic energy is proportional to the particle

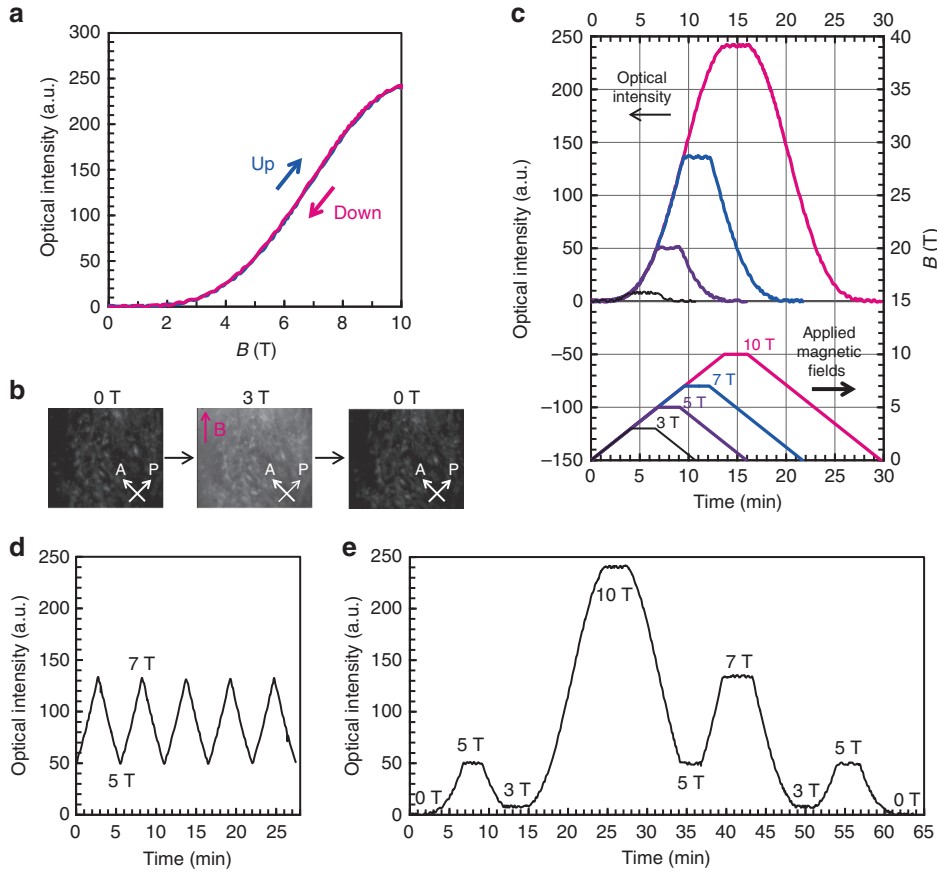

**Fig. 7** Magnetic modulation of light transmission in biphasic system. **a** Change in the intensity of light transmission through 8.5 vol% Iso/LC biphasic colloidal dispersion in 2-mm-thick optical cell, while magnetic field was increased from 0 T to 10 T and decreased from 10 T to 0 T. **b** In situ observation between crossed polarizers of Iso/LC biphasic colloidal dispersion when the magnetic field strength was changed between 0 T and 3 T. The magenta arrow represents the direction of the magnetic field. A Analyzer, P Polarizer. **c** Time course of transmitted light intensity of Iso/LC biphasic colloidal dispersion and the corresponding applied magnetic fields. **d**, **e** Modulation of light transmission through Iso/LC biphasic colloidal dispersion by tuning the strength of the applied magnetic field. Rate of change of magnetic field strength was 0.73 T min$^{-1}$

volume. For nano-sized materials, this energy is generally lower than the thermal energy, $kT$, resulting in randomization of the alignment caused by thermal fluctuations. The formation of LC states enables the HAp nanorods to align in magnetic fields because of the larger liquid crystal domains, which interact with the magnetic field[36,37]. POM observations with a wave plate (Supplementary Fig. 15) showed that the HAp nanorods are aligned perpendicular to the direction of the magnetic field, based on their positive birefringence (Supplementary Fig. 8). The alignment direction suggests that the liquid crystals in the present work have negative magnetic anisotropy[46]. For the magneto-responsive functions, the key discovery is the dynamic behaviors of Iso/LC biphasic colloidal dispersions. The faster optical switching in the biphasic system can be attributed to the Iso/LC biphasic nature of the system, with the LC phases dispersed in the form of droplets in the Iso aqueous phase. In the biphasic system, the directors of LC droplets are quickly oriented by magnetic torque upon application of magnetic fields and quickly relaxed by Brownian motion after removal of the magnetic fields (Fig. 8). We assume that the soft LC/aqueous interface of the LC droplets results in dynamic properties of the biphasic system in response to external magnetic fields, while dynamic magnetic response of the homogeneous system is disturbed by the anchoring effect at the LC/solid glass cell interface (Supplementary Fig. 11). The magnetic anisotropy of the LC droplets in the biphasic colloidal dispersion was estimated. The value of $\Delta\chi V$ was found to be of the order of $10^{-26}$ m$^3$, based on fitting of the theoretical order

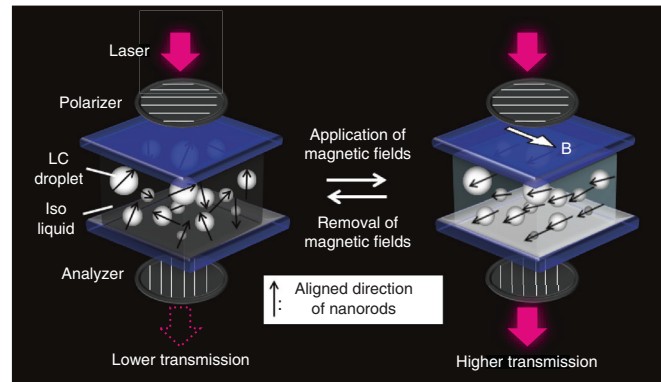

**Fig. 8** Dynamic optical switching in Iso/LC biphasic system in response to external magnetic fields. Schematic illustration of reversible optical switching behaviors of Iso/LC biphasic colloidal dispersion in response to changes in external magnetic field. In the biphasic system, the LC phase is dispersed in droplet form in the Iso aqueous phase

parameters under magnetic fields (Supplementary Fig. 16), where $\Delta\chi$ is the anisotropy of the magnetic susceptibility and $V$ is the domain size of the liquid crystal. The biphasic system contains droplets of various sizes, from nano to micron scales. If the average size is assumed to be 1 µm$^3$, the anisotropy of the magnetic susceptibility ($\Delta\chi$) is roughly calculated to be $-10^{-8}$. This

evaluation suggests that magneto-responsive functions can be achieved with small magnetic anisotropy by using the dynamic behavior of the biphasic system. There has been a focus on the observation[47–49] and simulation[50] of the formation of such biphasic states, but little attention has been paid to functionality. Here we shed new light on the functional properties of the biphasic state, which will advance the development of LC materials with dynamic functions. The liquid crystal can also be macroscopically oriented by a mechanical force. Control of the crystallization of HAp has been intensively studied for the development of structurally controlled nanomaterials inspired by bones and teeth[51–54], but macroscopic orientational control is still challenging in synthetic systems. LC HAp materials could provide platforms for structural control of HAp-based materials.

We developed a biomineralization-inspired method using PAA molecules for the synthesis of colloidal liquid crystals. Self-organization of HAp crystals with PAA molecules led to the formation of mesogenic nanorods with colloidal stability. Atomic-scale observations using aberration-corrected TEM showed the self-organized inorganic/organic hybrid structure. These HAp nanorods stabilized by PAA molecules maintain stable colloidal states over a few years without control of pH and temperature. The polymer additive is therefore responsible not only for morphological control, but also for efficient dispersion of the HAp nanorods, which is essential for LC ordering in colloidal states. In our synthetic approach, LC HAp nanorods were synthesized in the PAA concentration range 0.18–0.90 wt% (Supplementary Fig. 17). The aspect ratios and sizes of these mesogenic nanorods are similar. The nanorods synthesized at PAA concentrations lower than 0.54 wt% showed more viscous states in their condensed LC colloidal dispersions. The amounts of PAA molecules self-assembled with nanorods were decreased with decreasing PAA concentrations in the reaction solutions (Supplementary Fig. 18), which may decrease electrostatic repulsive forces between nanorods. These results suggest that appropriate PAA concentration range 0.54–0.90 wt% is important for the synthesis of LC HAp nanorods with dynamic properties. The present study suggest that bioinspired self-organization of HAp with PAA is an effective approach to the syntheses of these colloidal liquid crystals. It is expected that application of this method using self-organization of organic molecules with inorganic materials will expand the range of compounds available as liquid crystals. Because of features such as facile synthesis and the bio-friendliness of HAp, stimuli-responsive LC HAp is a promising bio-friendly functional material that could be used for optical devices, biosensors, artificial bones, dental implants, and cell culture scaffolds.

## Methods

**Preparation of HAp colloidal dispersions**. A 100 mM aqueous $CaCl_2$ solution containing PAA ($M_w = 2.0 \times 10^3$, $7.2 \times 10^{-1}$ wt%) was mixed with an equal volume of 100 mM aqueous $K_3PO_4$ solution. The mixed solution was stirred for 3 days at 60 °C and then centrifuged. The supernatant was decanted. Before drying, the collected precipitates were redispersed in deionized water. The dispersion was centrifuged, and the supernatant was removed. Finally, deionized water was added to the sediment to adjust the concentration. The composition of the final materials after drying was determined to be 82.46 wt% HAp, 14.4 wt% PAA molecules, and 3.14 wt% water molecules based on elemental analysis for C, H, and N elements. No impurities were detected in the EDS analyses (Supplementary Fig. 7).

**Characterization**. XRD patterns were recorded using a SmartLab (Rigaku) diffractometer with Cu Kα radiation. FTIR spectra were obtained with a JASCO/FTIR-660 Plus spectrometer using the KBr method. The Ca/P ratio of the HAp nanorods was determined using ICP/AES (Thermo Scientific, iCAP DUO-6300). PAA amounts in the final materials were analyzed using an Exeter Analytical CE-440 Elemental Analyser. TG measurements (Rigaku, TG-8120) were performed up to 1000 °C at a heating rate of 10 °C min⁻¹ under air flow (100 mL min⁻¹). The optical properties of the samples were investigated using a polarizing optical microscope (Olympus, BX51). The crystal morphologies were examined using SEM

(Hitachi, S-4700), with conductive treatment using a Hitachi E-1030 ion sputterer. TEM characterizations were performed using a conventional TEM (JEOL, JEM-2800) equipped with a window-less silicon drift director (Oxford Instruments, X-MaxN 100TLE), operated at 100 kV. The EDS analyses were conducted with a selected-area electron probe, for the supporting grid free HAp nanorod (Supplementary Fig. 7). HRTEM images were obtained using an aberration-corrected TEM (JEOL, JEM-ARM200F) at 120 kV. Given the 120-kV instrument with an information limit of 10 nm⁻¹, we determined the negative spherical aberration imaging condition[40]. The HRTEM image simulation was carried out by the xHREM software (HREM Research Inc.). DLS and zeta potential measurements were performed using a Zetasizer (Nano-ZS, from Malvern Instruments Ltd.). UV-vis transmittance spectra were recorded using a Jasco V-670 spectrometer equipped with an ISN-800T integrating sphere unit.

**Application of magnetic fields**. A magnetic field was applied at room temperature using a cryocooler-cooled superconducting magnet (Sumitomo Heavy Industries, Tokyo, Japan). The transmitted light intensity was recorded under crossed polarizers using a laser of wavelength 670 nm as the light source. The polarizer and analyzer were each set at an angle of 45° with respect to the direction of the applied magnetic field.

**Materials**. All chemical reagents used for the synthesis of HAp crystals were obtained from commercial sources. PAA ($M_w = 2.0 \times 10^3$) was purchased from Polysciences, Inc. (Warrington, PA, USA). $CaCl_2$ and $K_3PO_4$ were obtained from Wako Pure Chemicals Industries, Ltd. (Osaka, Japan). All reagents were used as received.

**Data availability**. The data that support the findings of this study are available from the corresponding author upon reasonable request.

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

## Acknowledgements

This study was partly supported by JSPS KAKENHI Grant Numbers JP22107003, JP15H02179, and JP17J09259. M.N. is grateful for financial support from a Japan Society for the Promotion of Science (JSPS) Research Fellowship for Young Scientists and the JSPS Program for Leading Graduate Schools (MERIT). The authors are grateful to Prof. Teruyuki Nagamune, Dr. Satoshi Yamaguchi, and Dr. Kosuke Minamihata for performing the zeta potential measurements. TEM observations were conducted at the Advanced Characterization Nanotechnology Platform at the University of Tokyo, which is supported by the "Nanotechnology Platform" of the Ministry of Education, Culture, Sports, Science and Technology (MEXT), Japan.

## Author contributions

T.K. conceived and designed the project. T.K. and M.N. wrote the paper. M.N. performed the experiments. S.K. and T.N. assisted with the experiments and data analysis. A.K. performed the TEM experiments. Y.I. advised on and discussed the TEM experiments. M.Y. designed the experiments for magnetic alignment. All authors read and commented on the manuscript.

## Additional information

**Competing interests:** The authors declare no competing financial interests.

