## [Peer Review File · Nature Communications]

Reviewers' comments:

Reviewer #1 (Remarks to the Author):

This manuscript describes the formation and properties of a new type of alignable hydroxyapatite-based composite material with liquid-crystalline (LC) properties. The authors show that the synthesis of hydroxyapatite in the presence of a poly(acrylic acid) (PAA) solution results in the formation of hydroxyapatite nanorods. Washing away most of the PAA from the mixture results in PAA-surface-stabilized/complexed hydroxyapatite nanorods that have LC properties because of the anisotropic shape of the hydroxyapatite. The authors show that these LC-based PAA-hydroxyapatite nanorods can be macroscopically aligned by mechanical shearing and magnetic fields to afford a new inorganic, bio-based LC material that might have some interesting materials applications.

To my knowledge, the formation of individual hydroxyapatite nanorods that undergo LC assembly and can be processed in this manner is unprecedented. The closest related work (see my comments/revisions below) is the formation of LC polymer/PAA/hydroxyapatite composites/blends by another group in which an added commercial LC polymer serves as the bulk matrix and is responsible for the LC behavior of the system. Given the novelty of the described new bio-based inorganic LC material and its potential relevance to biomaterials applications, this work should attract broad scientific interest. I recommend publication in Nature Communications after some minor revisions. Overall the manuscript is well-written, and the experiments and results collected are thorough and systematic. I only found a few minor items that need to be corrected or improved upon as listed below:

(1) A quick on-line literature search on "liquid crystal hydroxyapatite" showed one related piece of work: There is a recent paper in which another group prepared LC hydroxyapatite composites by adding organic polymers (including PAA) to hydroxyapatite nanocrystals: Shen, Deyong; Fang, Liming; Chen, Xiaofeng; Tang, Youhong, "Structure and properties of polyacrylic acid modified hydroxyapatite/liquid crystal polymer composite," Journal of Reinforced Plastics and Composites 2011, 30 (13), 1155-1163. This recent paper was not cited by the authors in their Introduction but should be, given that it appears to be related. The authors should also comment explicitly in the manuscript on how their current work differs from this prior work. It appears that the prior work blends PAA and a LC polymer to induce mesogenic properties in the hydroxyapatite-based composite, and that the LC polymer additive is really responsible for the LC properties of the mixture. The use of magnetic fields in the current work to align the authors' LC hydroxyapatite nanorods is clearly unprecedented, though.

(2) Page 2, lines 50–51: The authors only provide a very general justification for developing these new materials. Can the authors be more specific about what these new alignable hydroxyapatite-based colloidal LC mixtures might be potentially useful for or how they might be valuable? For example, as alignable starting materials/templates for making anisotropic synthetic bone- or teeth-like structural materials. Some potential applications for these new LC materials are listed at the very end of the manuscript, but it is better to include some mention of these at the beginning of the article in order to help justify the work upfront.

(3) Page 3, Results section, all reported data as average values and error bars: Error bars should only be reported to 1 sig fig. There is no value in reporting extra sig figs for error bars. Then, the decimal place precision of the reported measured avg data values should be truncated to that dictated by the 1 sig fig error bars.

(4) A listing of data on the purity and composition of the formed LC hydroxyapatite nanorods should

be included in the Structural Characterization section and/or Methods section. Although there is some composition data provided on the wt % PAA associated with the nanorods after washing in the main text and Suppl Info., those data are based on TGA mass loss (not direct element composition analysis). Some direct elemental analysis or EDAXS data on the final materials should be included in the main manuscript to establish nanorod purity and amount of PAA remaining.

(5) There are a small number of minor grammatical and formatting errors at various points in the main manuscript that should be corrected for clarity or style consistency. I have attached an annotated version of the PDF manuscript to return to the authors to help them identify and correct these minor errors.

Once these relatively minor revisions are taken care of, this manuscript should be publishable in Nature Communications.

Reviewer #2 (Remarks to the Author):

The manuscript reports an interesting study of synthesis of hydroxyapatite nanorods, which exhibit relatively high order parameter above critical concentrations that is analogous to lyotropic liquid crystals (LCs). It demonstrates that unidirectional shearing, spin-coating or external magnetic field all lead to a good alignment of the prepared hydroxyapatite nanorods. Furthermore, the authors report magneto-optical responsiveness of the LC hydroxyapatite, which enables modulation of light transmission by dynamic control of the LC order using external magnetic field. Overall, the work is significant, and should be published in Nature Communications. While the paper is strong in terms of characterization, I found it somewhat challenging to fully understand the interpretation of i) reasoning behind the synthetic methodology and ii) a number of measurements (see below):

1. In this work, the authors used 0.72wt% PAA in the synthesis process, resulting in 100 nm-in-length and 21 nm-in-width nanorods with aspect ratio of 5. The reason that the authors selected this critical dimension (and this particular concentration of PAA) is not clearly presented in this paper. As is well known, the phase behavior of colloidal particles strongly depends on their shape, especially, the aspect ratio. It would be desirable to either present a convincing explanation or make a clear comparison between the hydroxyapatite nanorods with different dimensions.

2. Inspection of Figure 6 reveals a twisting of the LC director within the droplets. Are the left and right kind of the observed twist equally probable? This observation is resembling of /analogous to the spatial confinement of achiral lyotropic chromonic LCs (e.g., DSCG tactoids), in which the observed twisting is due to the small twist constant, as compared to the elastic constants of splay and bend. A more detailed characterization and discussion would help to clarify the inter-colloidal interactions between the nanorods.

3. Page 7, line 220: "relaxation started immediately and it only took a few seconds to complete (Fig. 6d); this is 20 times faster than the relaxation speed of the homogeneous LC colloidal dispersions". This description is purely phenomenological. It would be more insightful if a single scaling argument to estimate the order of magnitude of relaxation time induced by rotational Brownian motion were used for both cases.

4. Page 9, Line 246: "In addition, the light transmission was 27 times higher at 10 T than at 3 T (Fig. 8c)". Why does the intensity of transmission light scale proportionally to B^3 ? Is there an underlying physical explanation? Would it be possible that such high magnetic field (10 T) imparts a change in

the anisotropy of the magnetic susceptibility or even domain size of the LC domains?

Overall this manuscript reports an interesting class of liquid crystals based on hydroxyapatite, but the manuscript needs some work for better interpretation and explanation of the results reported in this paper.

Recommendation: the paper merits publication in a revised form which should take into account the comments above

Reviewer #3 (Remarks to the Author):

The author reported a liquid crystal (LC) based on hydroxyapatite (HAp) nanorods. The HAp nanorods were synthesized using poly-acrylic acid (PAA) as the crystallization inhibitor. The alignment of the (LC) was tuned by applying external magnetic and mechanical fields. The atomic-scale structure of the product as well as its alignment behavior and magnetic-optic properties were thoroughly studied, and the results are convincing. To the best of the reviewer's knowledge, this is the first report of HAp-based LC. Moreover, tuning the orientation of HAp using magnetic field also has not been achieved before. However, following questions need to be clarified before the reviewer is able to correctly evaluate the impact of the work, and recommend it for publication in Nature Communications:

1. In line 58, the authors stated that "Hydroxyapatite is also known to be a diamagnetically anisotropic material", and cited the following literature: "31. Nakahira, A., Konishi, S., Nishimura, F., Iwasaka, M. & Ueno, S. Effect of a high magnetic field on the bioactivity of apatite-based biomaterials. *J. Appl. Phys.* 93, 8513–8515 (2003).". Later in line 181, this reference was mentioned again to support the "crystalline magnetic anisotropy". However, the reviewer is unable to find any evidence in this reference that could support the diamagnetical anisotropy of HAp crystal. In this reference, magnetic field was used to induce the nucleation of HAp, and "This may be related with magnetic influence on the balance of dissolution and precipitation. For example, it is likely that high magnetic fields induce the increase of supersaturation of various ions Ca^{2+} , HPO_4^{2-} , and PO_4^{3-} and the formation of many hydroxyapatite nuclei. Especially, the change of transport process due to the magnetic energy gradient on macroscale and the difference of ion concentration and also the effect of gradient magnetic fields could be given as the high magnetic field effect." (page 8515 of the reference), which is unrelated to the intrinsic property of HAp crystal. Therefore, there is actually neither any proper reference nor discussion/calculation in the present paper that could support the diamagnetical anisotropy of HAp.

In spite of this defect, the diamagnetism of HAp observed by the authors is not unexpected, since most materials possess diamagnetism, and it could be observed when the material is not paramagnetic/ferromagnetic. This includes most metals, salts and polymers. When the material is not spherical (e.g., rod-like), diamagnetical anisotropy will naturally happen due to the shape effect. However, this also means less novelty of the present finding. The authors need to provide more experimental/calculation/literature evidence to show that the diamagnetical anisotropy is indeed due to specific crystallographic structure of HAp crystals. Otherwise they should consider to change the corresponding expressions to prevent misleading.

2. In line 75 and 147, the author mentioned "removal of residual ions and PAA molecules" before the LC was formed, which is however ambiguous since the PAA molecules generated repulsive force and is essential for the formation of LC phase (described from line 138 to line 141). The authors need to clarify how the PAA was removed, and how much PAA is left in the LC, or make it clear if they actually only removed free PAA that was not incorporated with the HAp crystal.

3. According to the Method section, the authors first centrifuge the HAp nanorods out of the reaction (water) solution, then re-dispersed the nanorods back to water to form the LC. This implies the material is metastable and phase-separation probably will happen afterwards due to the higher density of HAp comparing with water. Could the authors provide any information about the stability of the material? Was the pH or temperature controlled to increase the stability of the material?

Other suggestions that might help to strengthen the manuscript include:

1. In line 194-195, it was mentioned that the nanorods are aligned perpendicular to the magnetic field direction, suggesting that the LC has negative magnetic anisotropy. References and discussions need to be added to clarify why that is.

2. The Result section should be mainly about the experimental results, without too much discussion/induction, while the Discussion section should be more that a summary/outlook. The authors should consider to separate the discussions in the Result section and move them into Discussion section, making them more clear and detailed, such as line 138-143, line 188-195, line 224-233 and Figure 7, line 253-259.

3. Line 43 and line 44-45 are repetitive.

4. Please use HAp or HAP as the abbreviation of hydroxyapatite.

5. The schematic illustration in Figure 3h is very similar to a SEM image and could be misleading. The authors should consider to plot it in the way similar to Figure 1 or Figure 7.

6. The references should be all after or before the punctuations.

7. The lengths above the scale bars (e.g., 100 nm) should be moved to the figure captions according to the requirement of Nature Communications.

Decision:

The referee invite the authors to revise their manuscript to address specific concerns before a final decision is reached. The diamagnetical anisotropy of HAp has to be properly discussed, and the experimental methods need to be more clearly described.

We thank the reviewers for giving us fruitful comments about our manuscript. We carefully revised our manuscript according to the comments. The responses to each reviewer are as follows.

Reviewer 1

Reviewer comments (General): This manuscript describes the formation and properties of a new type of alignable hydroxyapatite-based composite material with liquid-crystalline (LC) properties. The authors show that the synthesis of hydroxyapatite in the presence of a poly(acrylic acid) (PAA) solution results in the formation of hydroxyapatite nanorods. Washing away most of the PAA from the mixture results in PAA-surface-stabilized/complexed hydroxyapatite nanorods that have LC properties because of the anisotropic shape of the hydroxyapatite. The authors show that these LC-based PAA-hydroxyapatite nanorods can be macroscopically aligned by mechanical shearing and magnetic fields to afford a new inorganic, bio-based LC material that might have some interesting materials applications.

To my knowledge, the formation of individual hydroxyapatite nanorods that undergo LC assembly and can be processed in this manner is unprecedented. The closest related work (see my comments/revisions below) is the formation of LC polymer/PAA/hydroxyapatite composites/blends by another group in which an added commercial LC polymer serves as the bulk matrix and is responsible for the LC behavior of the system. Given the novelty of the described new bio-based inorganic LC material and its potential relevance to biomaterials applications, this work should attract broad scientific interest. I recommend publication in Nature Communications after some minor revisions.

Overall the manuscript is well-written, and the experiments and results collected are thorough and systematic. I only found a few minor items that need to be corrected or improved upon as listed below:

Author reply: We appreciate these positive comments. We revised our manuscript by taking account of your comments as follows.

[1] Reviewer comments: A quick on-line literature search on "liquid crystal hydroxyapatite" showed one related piece of work: There is a recent paper in which another group prepared LC hydroxyapatite composites by adding organic polymers (including PAA) to hydroxyapatite nanocrystals: Shen, Deyong; Fang, Liming; Chen, Xiaofeng; Tang, Youhong, "Structure and properties of polyacrylic acid modified hydroxyapatite/liquid crystal polymer composite," Journal of Reinforced Plastics and Composites 2011, 30 (13), 1155-1163. This recent paper was not cited by the authors in their Introduction but should be, given that it appears to be related. The authors should also comment explicitly in the manuscript on how their current work differs from this prior work. It appears that the prior work blends PAA and a LC polymer to induce mesogenic properties in the hydroxyapatite-based composite, and that the LC polymer additive is really responsible for the LC properties of the mixture. The use of magnetic fields in the current work to align the authors' LC hydroxyapatite nanorods is clearly unprecedented, though.

Author reply: We thank the kind suggestions. However, it does not seem that the suggested paper is suitable literature to be cited in our paper because the suggested paper does not report mesogenic properties of hydroxyapatite (HAp) crystals, but simple mixtures of LC polymer, HAp powder and PAA. The HAp crystals themselves show no liquid-crystalline properties. In contrast, our paper focus on synthesis of LC HAp nanocrystals and the alignment control using external stimuli. The research concepts and materials design are quite different and it is unfavorable to make a comparison of our work with the suggested paper.

[2] Reviewer comments: Page 2, lines 50–51: The authors only provide a very general justification for developing these new materials. Can the authors be more specific about what these new alignable hydroxyapatite-based colloidal LC mixtures might be potentially useful for or how they might be valuable? For example, as alignable starting materials/templates for making anisotropic synthetic bone- or teeth-like structural materials. Some potential applications for these new LC materials are listed at the very end of the manuscript, but it is better to include some mention of these at the beginning of the article in order to help justify the work upfront.

Author reply: We agree with these suggestions. For further justification of our work upfront, we revised the descriptions in Page 2, lines 8-11 as follows.

“If HAp could be endowed with LC properties, new bio-friendly and environmentally friendly functional materials could be developed, which might provide interesting materials applications including optical switching, biosensors and structural materials available for artificial bones, dental implants and cell culture scaffolds.”

[3] Reviewer comments: Page 3, Results section, all reported data as average values and error bars: Error bars should only be reported to 1 sig fig. There is no value in reporting extra sig figs for error bars. Then, the decimal place precision of the reported measured avg data values should be truncated to that dictated by the 1 sig fig error bars.

Author reply: We truncated all error bars of measured average values to 1 sig fig according to the PDF manuscript annotated by Reviewer 1.

[4] Reviewer comments: A listing of data on the purity and composition of the formed LC hydroxyapatite nanorods should be included in the Structural Characterization section and/or Methods section. Although there is some composition data provided on the wt % PAA associated with the nanorods after washing in the main text and Suppl Info., those data are based on TGA mass loss (not direct element composition analysis). Some direct elemental analysis or EDAXS data on the final materials should be included in the main manuscript to establish nanorod purity and amount of PAA remaining.

Author reply: We thank the valuable comments. According to these comments, we conducted elemental analyses of HAp nanorods using both of a CHN elemental analyzer and energy-dispersive X-ray spectroscopy (EDS). The elemental analyses for C, H and N elements revealed the presence of 14.4 wt% PAA molecules complexed with HAp nanorods, which is in good agreement with the TG measurements. In the EDS analyses, Ca, P, C and O elements were detected, all of which is ascribed to HAp $[\text{Ca}_{10}(\text{PO}_4)_6(\text{OH})_2]$, PAA $[(\text{C}_3\text{H}_4\text{O}_2)_n]$ or H_2O , while no impurities such as K^+ and Cl^- were included in a nanorod (Supplementary Fig. 7).

We added Supplementary Fig. 7 and descriptions in the Supplementary Information, In addition, we added descriptions about nanorod purity in Page 2, Lines 51-53 of the main text as follows.

“Energy-dispersive X-ray spectroscopy (EDS) analysis shows the presence of Ca, P, C and O elements included in a nanorod, which are all ascribed to HAp, PAA or water molecules, while no impurities such as K^+ and Cl^- were detected (Supplementary Fig. 7). These results suggest high purity of HAp nanorods.”

We added a description about the amount of PAA remaining in Page 3, Lines 15-17 of the main text as follows.

“The presence of PAA complexed with the nanorods was also confirmed by direct elemental analysis for C, H and N elements. The HAp/PAA ratio was estimated to be 5.7:1.0 by weight, which is in good agreement with the value estimated by the TG measurements.”

We added composition data based on elemental analyses in the Method Section (Page 5, Lines 31-33) as follows.

“The composition of the final materials after drying was determined to be 82.46 wt% HAp, 14.4 wt% PAA molecules and 3.14 wt% water molecules based on elemental analysis for C, H and N elements. No impurities were detected in the EDS analyses (Supplementary Fig. 7).”

We added explanation on the CHN elemental analyzer in the “Experimental section” (Page 5, Lines 37-38) as follows.

“PAA amounts in the final materials were analyzed using an Exeter Analytical CE-440 Elemental Analyser”

We added explanation on the EDS analyses in the “Experimental section” (Page 5, Line 41-44) as follows.

“TEM characterizations were performed using a conventional TEM (JEOL, JEM-2800) equipped with a window-less silicon drift detector (Oxford Instruments, X-MaxN 100TLE), operated at 100 kV. The EDS analyses were conducted with a selected-area electron probe, for the supporting grid free HAp nanorod (Supplementary Fig. 7).”

[5] Reviewer comments: There are a small number of minor grammatical and formatting errors at various points in the main manuscript that should be corrected for clarity or style consistency. I have attached an annotated version of the PDF manuscript to return to the authors to help them identify and correct these minor errors.

Author Reply: We thank the reviewer for the kind notes on the PDF manuscript. We revised those minor errors according to the suggestions.

Reviewer comments (Conclusion): Once these relatively minor revisions are taken care of, this manuscript should be publishable in Nature Communications.

Author reply: We believe that the manuscript revised based on the comments from the reviewers is now publishable in Nature Communications.

Reviewer 2

Reviewer comments (General): The manuscript reports an interesting study of synthesis of hydroxyapatite nanorods, which exhibit relatively high order parameter above critical concentrations that is analogous to lyotropic liquid crystals (LCs). It demonstrates that unidirectional shearing, spin-coating or external magnetic field all lead to a good alignment of the prepared hydroxyapatite nanorods. Furthermore, the authors report magneto-optical responsiveness of the LC hydroxyapatite, which enables modulation of light transmission by dynamic control of the LC order using external magnetic field. Overall, the work is significant, and should be published in Nature Communications. While the paper is strong in terms of characterization, I found it somewhat challenging to fully understand the interpretation of i) reasoning behind the synthetic methodology and ii) a number of measurements (see below):

Author reply: We appreciate these positive comments. We revised our manuscript by taking your comments into consideration as follows.

[1] **Reviewer comments:** In this work, the authors used 0.72wt% PAA in the synthesis process, resulting in 100 nm-in-length and 21 nm-in-width nanorods with aspect ratio of 5. The reason that the authors selected this critical dimension (and this particular concentration of PAA) is not clearly presented in this paper. As is well known, the phase behavior of colloidal particles strongly depends on their shape, especially, the aspect ratio. It would be desirable to either present a convincing explanation or make a clear comparison between the HAp nanorods with different dimensions.

Author reply: We thank the valuable comments. We examined the size and morphologies of hydroxyapatite (HAp) nanocrystals synthesized in various PAA concentrations with TEM observations. We found that mesogenic nanorods were obtained in the PAA concentration range 0.18–0.90 wt%. Their aspect ratios and dimensions are similar (Supplementary Fig. 17). The viscosities of LC colloidal dispersions of nanorods showed an increasing trend as the concentrations of PAA decrease. The amounts of PAA molecules self-assembled with nanorods were decreased

with decreasing PAA concentrations in the reaction solutions (Supplementary Fig. 18), which may decrease electrostatic repulsive forces between nanorods. To explain this point, we added Supplementary Figs. 17 and 18 and descriptions in the Supplementary Information. We also added descriptions in Page 5, Lines 11-17 of the main text as follows.

“In our synthetic approach, LC HAp nanorods were synthesized in the PAA concentration range 0.18–0.90 wt% (Supplementary Fig. 17). The aspect ratios and sizes of these mesogenic nanorods are similar. The nanorods synthesized at PAA concentrations lower than 0.54 wt% showed more viscous states in their condensed LC colloidal dispersions. The amounts of PAA molecules self-assembled with nanorods were decreased with decreasing PAA concentrations in the reaction solutions (Supplementary Fig. 18), which may decrease electrostatic repulsive forces between nanorods. These results suggest that appropriate PAA concentration range 0.54–0.90 wt% is important for the synthesis of LC HAp nanorods with dynamic properties.”

[2] **Reviewer comments:** Inspection of Figure 6 reveals a twisting of the LC director within the droplets. Are the left and right kind of the observed twist equally probable? This observation is resembling of /analogous to the spatial confinement of achiral lyotropic chromonic LCs (e.g., DSCG tactoids), in which the observed twisting is due to the small twist constant, as compared to the elastic constants of splay and bend. A more detailed characterization and discussion would help to clarify the inter-colloidal interactions between the nanorods.

Author reply: We thank the valuable comments. It is unlikely that the LC director within the droplets is twisted upon application of magnetic fields. Generally, the director of a LC domain is simply rotated by magnetic torque under magnetic fields. Such LC director rotation is reasonable in the case of LC droplets of HAp nanorods because the experimental results are well fitted by the theoretical calculation based on director orientation induced by magnetic torque (Supplementary Fig. 16b).

[3] **Reviewer comments:** Page 7, line 220: “relaxation started immediately and it only took a few seconds to complete (Fig. 6d); this is 20 times faster than the relaxation speed of the homogeneous LC colloidal dispersions”. This description is purely phenomenological. It would be more insightful if a single scaling argument to estimate the order of magnitude of relaxation time induced by rotational Brownian motion were used for both cases.

Author reply: In general, the rate of thermal relaxation induced by rotational Brownian motion depends on the viscosity of the liquid crystal at constant temperature. However, anchoring effects can also influence the relaxation dynamics of LC systems. Note that the homogeneous LC system of HAp nanorods shows hysteresis between the processes of magnetic response and the relaxation (Supplementary Fig. 11). This hysteresis suggests that HAp nanorods are anchored on the glass surface of optical cell and the dynamic orientation change is disturbed in the homogeneous system. In contrast, the biphasic system with LC domains dispersed in aqueous phase shows no hysteresis (Fig. 7a), suggesting the negligible anchoring effect. For these different systems, the difference of

relaxation time scales is not simply induced by rotational Brownian motion. We revised and added descriptions about the difference between the homogeneous and biphasic systems in Page 4, Lines 46-51 of the main text as follows.

“The faster optical switching in the biphasic system can be attributed to the Iso/LC biphasic nature of the system, with the LC phases dispersed in the form of droplets in the Iso aqueous phase. In the biphasic system, the directors of LC droplets are quickly oriented by magnetic torque upon application of magnetic fields and quickly relaxed by Brownian motion after removal of the magnetic fields (Fig. 8). We assume that the soft LC/aqueous interface of the LC droplets result in dynamic properties of the biphasic system in response to external magnetic fields, while dynamic magnetic response of the homogeneous system is disturbed by the anchoring effect at the LC/solid glass cell interface (Supplementary Fig. 11).”

[4] **Reviewer comments:** Page 9, Line 246: “In addition, the light transmission was 27 times higher at 10 T than at 3 T (Fig. 8c)”. Why does the intensity of transmission light scale proportionally to B^3 ? Is there an underlying physical explanation? Would it be possible that such high magnetic field (10 T) imparts a change in the anisotropy of the magnetic susceptibility or even domain size of the LC domains?

Author reply: The change of transmitted light intensity of liquid crystals results from change of order parameter based on LC director orientation induced by magnetic torque. In this model, the order parameter is described as a function of magnetic field strength, B , but the relation is complicated, as expressed by the equations (4)-(6) in Supplementary Fig. 16. Therefore, the intensity of light transmission scale is not simply proportional to B^3 as shown in Supplementary Fig. 16. Orientation change of HAp liquid crystals under magnetic fields are reasonably fitted by this theoretical calculation based on LC director orientation induced by magnetic torque (Supplementary Fig. 16b). Here anisotropy of the magnetic susceptibility and domain size of liquid crystals are assumed to be constant. We added a description in Page 4 Lines 27-28 of the main text as follows.

“The dependence of light transmission intensity on the field strength is in close agreement with the theoretical calculation based on orientation change of LC directors induced by magnetic torque (Supplementary Fig. 16).”

Reviewer comments (Conclusion): Recommendation: the paper merits publication in a revised form which should take into account the comments above

Author reply: We thank the positive recommendation. We believe that the manuscript revised by taking the comments from the reviewers into consideration is now publishable in Nature Communications.

Reviewer 3

Reviewer comments (General): The author reported a liquid crystal (LC) based on hydroxyapatite (HAp) nanorods. The HAp nanorods were synthesized using poly-acrylic acid (PAA) as the crystallization inhibitor. The alignment of the (LC) was tuned by applying external magnetic and mechanical fields. The atomic-scale structure of the product as well as its alignment behavior and magnetic-optic properties were thoroughly studied, and the results are convincing. To the best of the reviewer's knowledge, this is the first report of HAp-based LC. Moreover, tuning the orientation of HAp using magnetic field also has not been achieved before. However, following questions need to be clarified before the reviewer is able to correctly evaluate the impact of the work, and recommend it for publication in Nature Communications:

Author reply: We appreciate these positive comments. We revised our manuscript by taking account of your comments as follows.

[1] **Reviewer comments:** In line 58, the authors stated that "Hydroxyapatite is also known to be a diamagnetically anisotropic material", and cited the following literature: "31. Nakahira, A., Konishi, S., Nishimura, F., Iwasaka, M. & Ueno, S. Effect of a high magnetic field on the bioactivity of apatite-based biomaterials. J. Appl. Phys. 93, 8513–8515 (2003).". Later in line 181, this reference was mentioned again to support the "crystalline magnetic anisotropy". However, the reviewer is unable to find any evidence in this reference that could support the diamagnetical anisotropy of HAp crystal. In this reference, magnetic field was used to induce the nucleation of HAp, and "This may be related with magnetic influence on the balance of dissolution and precipitation. For example, it is likely that high magnetic fields induce the increase of supersaturation of various ions Ca^{2+} , HPO_4^{2-} , and PO_4^{3-} and the formation of many hydroxyapatite nuclei. Especially, the change of transport process due to the magnetic energy gradient on macroscale and the difference of ion concentration and also the effect of gradient magnetic fields could be given as the high magnetic field effect."(page 8515 of the reference), which is unrelated to the intrinsic property of HAp crystal. Therefore, there is actually neither any proper reference nor discussion/calculation in the present paper that could support the diamagnetical anisotropy of HAp.

In spite of this defect, the diamagnetism of HAp observed by the authors is not unexpected, since most materials possess diamagnetism, and it could be observed when the material is not paramagnetic/ferromagnetic. This includes most metals, salts and polymers. When the material is not spherical (e.g., rod-like), diamagnetical anisotropy will naturally happen due to the shape effect. However, this also means less novelty of the present finding. The authors need to provide more experimental/calculation/literature evidence to show that the diamagnetical anisotropy is indeed due to specific crystallographic structure of HAp crystals. Otherwise they should consider to change the corresponding expressions to prevent misleading.

Author reply: We agree with these reviewer's comments that the reference does not include reasonable evidence for diamagnetic magnetocrystalline anisotropy of HAp. Alternatively, we cited

the following two references, which experimentally support the magnetocrystalline anisotropy of HAp.

“31. Inoue, K., Sassa, K., Yokogawa, Y., Sakka, Y., Okido, M. & Asai S. Control of crystal orientation of hydroxyapatite by imposition of a high magnetic field" *Mater. Trans.* **44**, 1133–1137 (2003).”

“32. Akiyama, J. *et al.* Formation of c-axis aligned polycrystal hydroxyapatite using high magnetic field with mechanical sample rotation" *Mater. Trans.* **46**, 203–206 (2005).”

These papers report that *c* axes of HAp crystals align perpendicular to magnetic field directions, and experimentally demonstrate that HAp is a magnetically anisotropic material with a smaller magnetic susceptibility in the *c* axis compared with *a* or *b* axis. These are consistent with our experimental results that the *c* axes of LC HAp nanorods aligned perpendicular to magnetic field directions (Supplementary Fig. 15).

In addition, even if shape effect causes magnetic alignment of rod-like particles, it is energetically favorable to align their long axes parallel to the direction of magnetic fields [References: “Zhang, S. *et al.* Liquid crystalline order and magnetocrystalline anisotropy in magnetically doped semiconducting ZnO nanowires. *ACS Nano* **5**, 8357–8364 (2011). (Page 8362)” and “Leferink Op Reinink, A. B. G. M. *et al.* Phase behaviour of lyotropic liquid crystals in external fields and confinement. *Eur. Phys. J. Spec. Top.* **222**, 3053–3069 (2013). (Page 3059, Lines 3-8)”].

In the case of LC HAp nanorods, their long axes are aligned perpendicular to the magnetic field directions.

These literatures and our experimental results suggest that magnetocrystalline anisotropy of HAp crystals mainly contributed to the magnetic alignment of the HAp liquid crystals.

[2] **Reviewer comments:** In line 75 and 147, the author mentioned “removal of residual ions and PAA molecules” before the LC was formed, which is however ambiguous since the PAA molecules generated repulsive force and is essential for the formation of LC phase (described from line 138 to line 141). The authors need to clarify how the PAA was removed, and how much PAA is left in the LC, or make it clear if they actually only removed free PAA that was not incorporated with the HAp crystal.

Author reply: In the purification process, we just removed free inorganic ions that were not incorporated in HAp crystals and free PAA molecules that were not hybridized with the HAp crystals. We revised the description in the legend of Figure 1 and in Page 3, Lines 22-23 of the main text as follows.

“removal of residual inorganic ions and free PAA molecules that were not complexed with HAp crystals”

[3] **Reviewer comments:** According to the Method section, the authors first centrifuge the HAp nanorods out of the reaction (water) solution, then re-dispersed the nanorods back to water to form the LC. This implies the material is metastable and phase-separation probably will happen afterwards due to the higher density of HAp comparing with water. Could the authors provide any

information about the stability of the material? Was the pH or temperature controlled to increase the stability of the material?

Author reply: The aqueous colloidal dispersions of HAp nanorods are highly stable and no sedimentation and aggregation were observed within a few years without control of pH and temperature. We added a description on the colloidal stability in Page 5, Lines 9-10 of the main text as follows.

“These HAp nanorods stabilized by PAA molecules maintain stable colloidal states over a few years without control of pH and temperature.”

Other suggestions from Reviewer 3

[1] **Reviewer comments:** In line 194-195, it was mentioned that the nanorods are aligned perpendicular to the magnetic field direction, suggesting that the LC has negative magnetic anisotropy. References and discussions need to be added to clarify why that is.

Author reply: To address these comments, we cited the following reference that reports negative magnetic anisotropy of LC cellulose whiskers, which align perpendicular to the direction of applied magnetic fields.

“43. Kimura, F., Kimura, T., Tamura, M., Hirai, A., Ikuno M. & Horii, F. Magnetic alignment of the chiral nematic phase of a cellulose microfibril suspension. *Langmuir* **21**, 2034–2037 (2005).”

[2] **Reviewer comments:** The Result section should be mainly about the experimental results, without too much discussion/induction, while the Discussion section should be more that a summary/outlook. The authors should consider to separate the discussions in the Result section and move them into Discussion section, making them more clear and detailed, such as line 138-143, line 188-195, line 224-233 and Figure 7, line 253-259.

Author reply: According to these suggestions, we moved those descriptions and the figure into Discussion section.

[3] **Reviewer comments:** Line 43 and line 44-45 are repetitive.

Author reply: According to the comments, we revised the sentences in Page 2, Line 1-2 as follows. “Liquid crystals form fluid but ordered molecular assemblies¹⁻³ and these self-assembled structures can be dynamically controlled by application of external stimuli such as electric and magnetic fields, and mechanical forces, leading to various functions⁴⁻¹¹.”

[4] **Reviewer comments:** Please use HAp or HAP as the abbreviation of hydroxyapatite.

Author reply: According to the comments, we use “HAp” as the abbreviation of hydroxyapatite throughout the revised manuscript.

[5] Reviewer comments: The schematic illustration in Figure 3h is very similar to a SEM image and could be misleading. The authors should consider to plot it in the way similar to Figure 1 or Figure 7.

Author reply: To avoid such misinterpretation, we changed the background color of Figure 3h.

[6] Reviewer comments: The references should be all after or before the punctuations.

Author reply: We thank the reviewer for the comments. We moved all reference numbers before punctuations.

[7] Reviewer comments: The lengths above the scale bars (e.g., 100 nm) should be moved to the figure captions according to the requirement of Nature Communications.

Author reply: We thank the reviewer for the comments. We moved all the labels of scale bars on the figures to the figure captions.

Reviewer comments (Conclusion): Decision: The referee invite the authors to revise their manuscript to address specific concerns before a final decision is reached. The diamagnetical anisotropy of HAp has to be properly discussed, and the experimental methods need to be more clearly described.

Author reply: We carefully revised our manuscript by taking account of your comments on magnetic anisotropy of HAp and the experimental methods, and the other suggestions. We believe that the revised manuscript is now publishable in Nature Communications.

Revisions based on Manuscript Checklist

According to the checklist for abstract, we added short descriptions of background in the first part of Abstract section as follows.

“Liquid crystals are mostly formed by self-assembly of organic molecules. In contrast, inorganic materials available as liquid crystals are limited.”

Reviewers' comments:

Reviewer #1 (Remarks to the Author):

I think that this revised manuscript has adequately addressed pretty much all of my suggestions and criticisms from my original review.

The only thing that I recommend as a minor revision is that the authors do include a specific footnote/ref in the revised manuscript that the Zhen et al. paper on polyacrylic acid modified hydroxyapatite/liquid crystal polymer composites that I noted previously (Journal of Reinforced Plastics and Composites 2011, 30 (13), 1155-1163) that states that this prior paper "does not report mesogenic properties of hydroxyapatite (HAp) crystals, but simple mixtures of LC polymer, HAp powder and PAA, as stated in the authors' rebuttal letter", as stated in the rebuttal letter. Since there are very few papers on LC hydroxyapatite materials that can be found in the literature, I think it is good for the readers to know how the current paper is different from any papers on ca. the same topic. Then, it does not look like the authors are being selective about citing other work without a reason.

This is a minor point that I will defer to the editor's decision to include or not. Otherwise, I recommend publication of the revised manuscript on all other points.

Reviewer #2 (Remarks to the Author):

I have read the response of the authors to the first round of reviews. I have also read the revised manuscript. The revised manuscript is substantially improved in terms of articulation of a technical advance (synthetic methodology and characterization of the hydroxyapatite nanorods). The paper is also improved in terms of the quality of writing. I am pleased to be able to support its publication.

Reviewer #3 (Remarks to the Author):

I am very glad to see that most of the comments from me as well as other reviewers have been well taken care of, and the quality of the manuscript has been significantly improved. However, I still could not fully agree with the authors' response to my comment 1). In the two papers listed by the authors:

"31. Inoue, K., Sassa, K., Yokogawa, Y., Sakka, Y., Okido, M. & Asai S. Control of crystal orientation of hydroxyapatite by imposition of a high magnetic field" Mater. Trans. 44, 1133–1137 (2003)."

"32. Akiyama, J. et al. Formation of c-axis aligned polycrystal hydroxyapatite using high magnetic field with mechanical sample rotation" Mater. Trans. 46, 203–206 (2005)."

HAp was not described as a crystal with diamagnetism anisotropy. Instead, according to these two papers, HAp is a non-magnetic material with lower magnetic susceptibility in c-axis comparing with a- and b-axis. These two papers are useful in supporting that orientation of HAp crystals could be tuned by strong magnetic field. However, they are unable to support that "HAp is also known to be a diamagnetically anisotropic material" (line 59).

Still, as I have mentioned in the previous comment, the HAp/pAA nanorods reported here do behave like polymer fibers that show diamagnetic anisotropy. This is nicely shown in another paper mentioned by the authors:

"43. Kimura, F., Kimura, T., Tamura, M., Hirai, A., Ikuno M. & Horii, F. Magnetic alignment of the chiral nematic phase of a cellulose microfibril suspension. *Langmuir* 21, 2034–2037 (2005)."
Since the HAp/pAA nanorods are diamagnetic, they have to align their long axis perpendicular to the magnetic field direction to minimize the magnetic energy due to shape effect, which is different from paramagnetic nanorods (such as Co-doped ZnO reported in "Zhang, S. et al. Liquid crystalline order and magnetocrystalline anisotropy in magnetically doped semiconducting ZnO nanowires. *ACS Nano* 5, 8357–8364 (2011). (Page 8362)") that will align the long axis parallel to the magnetic field to minimize the magnetic energy.

It seems that there still lacks a conclusive report/experiment about the diamagnetic anisotropy of HAp crystal. However, this does not really affect the conclusion of current manuscript. I would suggest the authors to revise the corresponding paragraph as follows in order to prevent misleading:

Line 59-63:

HAp is also known to be a diamagnetically anisotropic material^{31,32} and can potentially be oriented in magnetic fields. However, to the best of our knowledge, the alignment of HAp nanocrystals by magnetic fields has not been achieved. For LC HAp, the use of magnetic fields could be promising for orientational control because self-assembly into LC ordered fluids can promote magnetic alignment³³⁻³⁵. Our aim was to functionalize HAp liquid crystals through dynamic control of the LC order by application of external magnetic fields.

Into:

High external magnetic field has been instrumental in orienting diamagnetic or paramagnetic materials with small anisotropic magnetic susceptibility, such as alumina, titania or HAp.^{31,32}(add also these two references: "1.Suzuki T S, Sakka Y. Fabrication of textured titania by slip casting in a high magnetic field followed by heating[J]. *Japanese Journal of Applied Physics*, 2002, 41(11A): L1272." 2. "Sakka Y, Suzuki T S, Tanabe N, et al. Alignment of titania whisker by colloidal filtration in a high magnetic field[J]. *Japanese journal of applied physics*, 2002, 41(12A): L1416.") For LC HAp, the use of magnetic fields could be promising for orientational control because self-assembly into LC ordered fluids can promote magnetic alignment³³⁻³⁵. Our aim was to functionalize HAp liquid crystals through dynamic control of the LC order by application of external magnetic fields.

After this minor revision, I think this manuscript should merit publication in *Nature Communications*.

Conclusion: The reviewer recommend this manuscript to be published in *Nature Communications* after minor revisions

We thank the reviewers for giving us valuable comments about our manuscript NCOMMS-17-16457A in the second round of reviews. We revised our manuscript according to the comments. All changes are highlighted by yellow marks in the manuscript. The responses to each reviewer are as follows.

Reviewer 1

Reviewer comments (General): I think that this revised manuscript has adequately addressed pretty much all of my suggestions and criticisms from my original review.

Author reply: We appreciate these positive comments. We revised our manuscript according to your comments as follows.

Reviewer comments: The only thing that I recommend as a minor revision is that the authors do include a specific footnote/ref in the revised manuscript that the Zhen et al. paper on polyacrylic acid modified hydroxyapatite/liquid crystal polymer composites that I noted previously (Journal of Reinforced Plastics and Composites 2011, 30 (13), 1155-1163) that states that this prior paper "does not report mesogenic properties of hydroxyapatite (HAp) crystals, but simple mixtures of LC polymer, HAp powder and PAA, as stated in the authors' rebuttal letter", as stated in the rebuttal letter. Since there are very few papers on LC hydroxyapatite materials that can be found in the literature, I think it is good for the readers to know how the current paper is different from any papers on ca. the same topic. Then, it does not look like the authors are being selective about citing other work without a reason.

Author reply: According to these comments, we revised the descriptions in Page 2, lines 7–10, and added the reference [24. Shen, D., Fang, L., Chen, X. & Tang, Y. Structure and properties of polyacrylic acid modified hydroxyapatite/liquid crystal polymer composite. *J. Reinf. Plast. Comp.* **30**, 1155–1163 (2011).] as follows.

“As for combination of liquid crystals and HAp crystals, a simple mixture of a LC thermotropic aromatic polyester and non-LC HAp particles was previously reported²⁴. However, to the best of our knowledge, there was no report on HAp crystals which themselves exhibit liquid crystallinity.”

Reviewer comments (Conclusion): This is a minor point that I will defer to the editor's decision to include or not. Otherwise, I recommend publication of the revised manuscript on all other points.

Author reply: We believe that the manuscript revised based on the comments is now ready for publication in Nature Communications.

Reviewer 2

Reviewer comments (Conclusion): I have read the response of the authors to the first round of reviews. I have also read the revised manuscript. The revised manuscript is substantially improved in terms of articulation of a technical advance (synthetic methodology and characterization of the hydroxyapatite nanorods). The paper is also improved in terms of the quality of writing. I am pleased to be able to support its publication.

Author reply: We appreciate the support for publication of our manuscript in Nature Communications.

Reviewer 3

Reviewer comments (General): I am very glad to see that most of the comments from me as well as other reviewers have been well taken care of, and the quality of the manuscript has been significantly improved. However, I still could not fully agree with the authors' response to my comment 1.

Author reply: We revised the manuscript according to your comments as follows.

Reviewer comments: In the two papers listed by the authors:

“31. Inoue, K., Sassa, K., Yokogawa, Y., Sakka, Y., Okido, M. & Asai S. Control of crystal orientation of hydroxyapatite by imposition of a high magnetic field" Mater. Trans. 44, 1133–1137 (2003).”

“32. Akiyama, J. et al. Formation of c-axis aligned polycrystal hydroxyapatite using high magnetic field with mechanical sample rotation" Mater. Trans. 46, 203–206 (2005).”

HAp was not described as a crystal with diamagnetism anisotropy. Instead, according to these two papers, HAp is a non-magnetic material with lower magnetic susceptibility in c-axis comparing with a- and b-axis. These two papers are useful in supporting that orientation of HAp crystals could be tuned by strong magnetic field. However, they are unable to support that “HAp is also known to be a diamagnetically anisotropic material” (line 59).

Still, as I have mentioned in the previous comment, the HAp/pAA nanorods reported here do behave like polymer fibers that show diamagnetic anisotropy. This is nicely shown in another paper mentioned by the authors:

“43. Kimura, F., Kimura, T., Tamura, M., Hirai, A., Ikuno M. & Horii, F. Magnetic alignment of the chiral nematic phase of a cellulose microfibril suspension. Langmuir 21, 2034–2037 (2005).”

Since the HAp/pAA nanorods are diamagnetic, they have to align their long axis perpendicular to the magnetic field direction to minimize the magnetic energy due to shape effect, which is different from paramagnetic nanorods (such as Co-doped ZnO reported in “Zhang, S. et al. Liquid crystalline order and magnetocrystalline anisotropy in magnetically doped semiconducting ZnO nanowires. *ACS Nano* 5, 8357–8364 (2011). (Page 8362)”) that will align the long axis parallel to the magnetic field to minimize the magnetic energy.

It seems that there still lacks a conclusive report/experiment about the diamagnetic anisotropy of HAp crystal. However, this does not really affect the conclusion of current manuscript. I would suggest the authors to revise the corresponding paragraph as follows in order to prevent misleading:

Line 59-63:

HAp is also known to be a diamagnetically anisotropic material^{31,32} and can potentially be oriented in magnetic fields. However, to the best of our knowledge, the alignment of HAp nanocrystals by magnetic fields has not been achieved. For LC HAp, the use of magnetic fields could be promising for orientational control because self-assembly into LC ordered fluids can promote magnetic alignment³³⁻³⁵. Our aim was to functionalize HAp liquid crystals through dynamic control of the LC order by application of external magnetic fields.

Into:

High external magnetic field has been instrumental in orienting diamagnetic or paramagnetic materials with small anisotropic magnetic susceptibility, such as alumina, titania or HAp.^{31,32}(add also these two references: “1.Suzuki T S, Sakka Y. Fabrication of textured titania by slip casting in a high magnetic field followed by heating[J]. *Japanese Journal of Applied Physics*, 2002, 41(11A): L1272.” 2. “Sakka Y, Suzuki T S, Tanabe N, et al. Alignment of titania whisker by colloidal filtration in a high magnetic field[J]. *Japanese journal of applied physics*, 2002, 41(12A): L1416.”) For LC HAp, the use of magnetic fields could be promising for orientational control because self-assembly into LC ordered fluids can promote magnetic alignment³³⁻³⁵. Our aim was to functionalize HAp liquid crystals through dynamic control of the LC order by application of external magnetic fields.

Author reply: We agree with these comments. According to the comments, we revised the descriptions in Page 2, lines 19–22, and added two additional references [34. Suzuki, T. S. & Sakka, Y. Fabrication of textured titania by slip casting in a high magnetic field followed by heating. *Jpn. J. Appl. Phys.* **41**, L1272 (2002). 35. Sakka, Y., Suzuki, T. S., Tanabe, N., Asai, S. & Kitazawa, K. Alignment of titania whisker by colloidal filtration in a high magnetic field. *Jpn. J. Appl. Phys.* **41**, L1416 (2002).] as follows.

“High external magnetic field has been instrumental in orienting diamagnetic or paramagnetic materials with small anisotropic magnetic susceptibility, such as alumina, titania or HAp³²⁻³⁵. For LC HAp, the use of magnetic fields could be promising for orientational control because self-assembly

into LC ordered fluids can promote magnetic alignment³⁶⁻³⁸. Our aim was to functionalize HAp liquid crystals through dynamic control of the LC order by application of external magnetic fields.”

Reviewer comments (Conclusion): After this minor revision, I think this manuscript should merit publication in Nature Communications.

Author reply: We appreciate these positive comments. We believe that the manuscript revised according to the comments is now publishable in Nature Communications.